# Dual Regime Mode-Locked and Q-Switched Erbium-Doped Fiber Laser by Employing Graphene Filament–Chitin Film-Based Passive Saturable Absorber

**DOI:** 10.3390/mi14051048

**Published:** 2023-05-14

**Authors:** Siti Nur Fatin Zuikafly, Harith Ahmad, Mohd Faizal Ismail, Mohd Azizi Abdul Rahman, Wira Jazair Yahya, Nurulakmar Abu Husain, Khairil Anwar Abu Kassim, Hafizal Yahaya, Fauzan Ahmad

**Affiliations:** 1Malaysia-Japan International Institute of Technology, Universiti Teknologi Malaysia, Kuala Lumpur 54100, Malaysia; sn.fatin@utm.my (S.N.F.Z.); azizi.kl@utm.my (M.A.A.R.); wira@utm.my (W.J.Y.); hafizal.kl@utm.my (H.Y.); 2Photonics Research Centre, University of Malaya, Kuala Lumpur 50603, Malaysia; harith@um.edu.my (H.A.); faizalis@um.edu.my (M.F.I.); 3Malaysian Institute of Road Safety Research, Kajang 43000, Malaysia; khairilanwar@miros.gov.my

**Keywords:** graphene, chitin, fiber laser, mode-locking, q-switching

## Abstract

We investigate the dynamics of high energy dual regime unidirectional Erbium-doped fiber laser in ring cavity, which is passively Q-switched and mode-locked through the use of an environmentally friendly graphene filament–chitin film-based saturable absorber. The graphene–chitin passive saturable absorber allows the option for different operating regimes of the laser by simple adjustment of the input pump power, yielding, simultaneously, highly stable and high energy Q-switched pulses at 82.08 nJ and 1.08 ps mode-locked pulses. The finding can have applications in a multitude of fields due to its versatility and the regime of operation that is on demand.

## 1. Introduction

The 1.5 µm band is best known as the standard telecommunication band, of which the losses in the silica fiber would be minimal compared to shorter or longer wavelength bands. Mutually, mode-locking operations in this region play an irreplaceable role in commercial and non-commercial fields, including in metrology and holography [1,2,3]. The active approach to achieving a phase-locked pulse in a laser uses an external modulator that aids in modulating the intracavity losses, involving more components and compromising the flexibility and robustness of the laser [4]. Alternatively, the passive approach uses the nonlinearity of a saturable absorber (SA) through real optical materials with nonlinear absorption characteristics, making it simpler and more flexible/robust in design. SAs have seen usage in the generation of either passive Q-switching or mode-locking, but rarely do both happen simultaneously in one laser operation. Neither are to be confused with Q-switched mode-locking, which often leads to unstable lasing operation (also known as Q-switching instabilities), hence its lack of use in applications, and simultaneous Q-switching and mode-locking not only offer regime switchability but also provide the option of the application of both whenever necessary. Since Q-switching yields a larger pulse energy while mode-locking produces narrower pulse width, applications can utilize the advantage of both with just a simple adjustment. Some researchers have reported Q-switched and mode-locked lasing, but this is often when using a combination of both passive SA and an intracavity polarization device to switch between the regimes [5,6,7,8,9]. Another Q-switched and mode-locked erbium-doped fiber laser has been reported using gadolinium oxide-based passive SA, although it requires cavity length adjustment in order to obtain stable phase-locked lasing [10]. Some SAs have also been reported to produce both Q-switched and mode-locked lasers using this method, but with significantly reduced frequency [7,11,12,13]. This work, interestingly, will report on the sole use of a passive SA inside a simpler cavity device, with no involvement of any polarization controller or cavity adjustment to produce a Q-switched and mode-locked laser simultaneously.

Two-dimensional (2D) layered materials are among the most well exploited SAs, owing to their high nonlinearity, high saturable absorption, and high damage threshold, with graphene and carbon nanotubes (CNTs) still spearheading the generation of ultrashort pulses due to their performance reproducibility and their facile fabrication. Their sub-picosecond relaxation time is beneficial because it makes them a reliable alternative to the conventional use of a semiconductor saturable absorber mirror (SESAM), which requires a complicated post-fabrication process. Compared to CNT, graphene works as a fast SA owing to its relaxation time of approximately 100–200 fs, and, thus, is more capable of having high a mode-locking pump power threshold. Additionally, there is a limitation on the working wavelength of the diameter-dependent CNTs. Having an independent wavelength absorption also means that graphene resonantly absorbs light [10] regardless of the wavelength range, which makes it more versatile and more flexible in multiple fields of applications.

Although graphene has been extensively exploited in terms of its performance parameters in the years that follow its first demonstration as SA in 2009 [14], such as the maximum repetition rate, shortest pulse width, polarization independent mode-locking, etc., and although its developments or related new findings congested somewhat after 2015, the optimization and alternatives to its fabrication are still vital to obtain an ideal mode-locked device and explore greener alternatives in order to achieve all-fiber laser mode-locking operations. It is worth noting that graphene is the pioneer to many of the emerging SAs, too, and still stands out today due to its superiority and its versatility [15,16]. To make incorporating SA devices in fiber laser cavities easier, regardless of the base material, researchers have usually opted for host polymers, such as polyvinyl alcohol (PVA) and polyethylene oxide (PEO) [17,18,19,20]. In order not to be limited by synthetic sources, the synthesis of graphene film in this current work made use of chitin in an attempt to achieve cost effectiveness and production and to also yield scalability. Produced from natural resources, such as plants and crustacean shells, chitin is the second most abundant natural product that has proved feasible to manufacture at a low cost. In telecommunication applications, especially, a polymer with C-F overtones, such as chitin, with low absorption losses at the desired wavelength, is better in terms of stability [21].

We have reported the successful implementation of a biodegradable and biocompatible chitin in a graphene-based SA device in the eye-safe region [22], and this paper will explore both its Q-switching and mode-locking ability in the telecommunication region, relying on the SA’s low saturation intensity and high thermal damage resistance with no cavity adjustment. This study can provide benefits to the application of light detection and ranging (LIDAR), in which short pulses of light allow for a very precise distance measurement, especially in compact and lightweight LIDAR systems, such as autonomous vehicles, drones, and robots. Additionally, pulsed fiber lasers, which are highly reliable with low noise in the telecommunication field (as offered by an EDFL), are well-suited to harsh and remote environments.

## 2. Materials, Methods and Characterizations

The graphene–chitin film used in this study was fabricated following the same procedure as reported in [22,23]. The graphene used in this study was yielded from a 3D printer filament obtained online from blackmagic3d. The filament of diameter 1.75 mm has a volume resistivity of 1 ohm-cm. Through a 3D printer nozzle at 210 °C, the filament was extruded with a resulting diameter of 400 µm, making it dissolve easier in the following step. A total of 25 mg of the filament was mixed with 1 mL of tetrahydrofuran (THF), producing a graphene-THF suspension upon vigorous ultrasonic mixing. Chitin, on the other hand, was produced separately. Fresh oyster mushrooms (*Pleurotus ostreatus*) were extracted to produce the chitin and 42 g of nanofibers were successfully extracted from 3 kg of whole mushrooms. Initially, the mushrooms were blended for 5 min using a regular domestic blender before undergoing hot water extraction to remove any water-soluble components. Following that, the *Pleurotus ostreatus* tissues underwent alkaline deproteinization treatment (1M NaOH) at 65 °C for 3 h. This is effective at removing proteins, lipids, and alkali-soluble glucan. Water was then added to the slurry in order to increase its volume to 1.5 l. The suspension was stirred for 30 min at 85 °C. Through 15 min of centrifugation (ThermoScientific, Sorvall Legend RT^+^), at 7000 rpm, the soluble components and excess water were removed to produce precipitate (cake), which was then soaked in alkaline solution. The suspension was heated to 65 °C for 3 h while stirring continuously before it was neutralized by re-centrifugation in excess water. The neutralized cake was then re-suspended in water (1:40 *w*/*v* ratio) and dispersed by final blending for another 1 min. The suspension was stored at 4 °C until further use. The produced graphene-THF suspension and chitin was mixed together with a one-to-one (1:1) ratio, totalling up to 5 mL. A 1-h ultrasonication process was then conducted to break down any agglomeration and to stack graphene (since graphene is known to have a strong Van der Waal cohesive force), producing a well-dispersed graphene in chitin. A thin yet sturdy film was obtained after a 36-h ambient temperature drying in a petri dish, revealing an SA with 50 µm thickness, as measured using a 3D laser microscope. Sturdy film was obtained after a 36 h ambient temperature drying process in a petri dish.

The graphene–chitin film was first characterized using field emission scanning electron microscope (FESEM) and Raman spectroscopy for their physical morphologies and signature peak profiles, respectively, as shown in Figure 1, from which the flaky and fibrous structure of chitin are clearly visible among the evenly distributed graphene itself. As compared to the free-standing chitin film, the graphene–chitin film was visibly less flakey, an indicator of mechanical improvement, as many has reported before [24,25]. While chitin did not exhibit any peak upon the excitation in Raman spectroscopy, both graphene and graphene-chitin matched well with the specified Raman peak profiles for graphene [26], which indicates the successful incorporation of graphene in the host polymer [27,28]. The calculated G/2D intensity ratio of less than 2 indicates that the graphene is a multilayer graphene.

A twin-balanced detector, as illustrated in Figure 2a, was then used to measure the nonlinear optical response of the graphene–chitin film, of which the mode-locking was sourced through a 1.5 µm Elmo Femtosecond Erbium Laser (MenloSystems) with a pulse width and a repetition rate of ˂150 fs and 100 Mhz, respectively. Figure 2 shows the modulation depth of approximately 15.08%, which was measured using a saturable absorption model [29]––A value considered high for the case of multi-layered graphene, which is known to have a low modulation depth [30,31]. The saturation intensity was measured at 0.01 MW/cm^2^ with a saturable loss of around 85%.

The transmission and absorption of the graphene–chitin film was recorded at 39% and 42%, respectively, in the 1500 nm region, as shown in Figure 3a,b. Visible transmittance can be observed from 1000 nm to 2000 nm thanks to the zero-bandgap property of graphene, which contributed to its wideband absorption ability. The steady and continuous absorption of at least 42% throughout the near infrared region was higher than several reported absorptions of graphene based SAs [32,33]. The Tauc plot, as shown in Figure 4, shows that the section of the straight line meets at zero absorption coefficient and photon energy, concurring to the theoretical value of 0.289 eV of graphene electronic band.

Additionally, the thickness of the graphene–chitin film was investigated and found to be approximately 50 μm, as shown in Figure 5. Acting as a fast optical switch that can rapidly change its absorption properties in response to changes in the incident optical power, the thickness of the SA film can affect the threshold power required for the SA to saturate as well as the duration of the Q-switched pulse itself, which can be longer with thicker film. This, however, can be affected by other factors, including the cavity length, the pump power, and the SA material itself. The chemical analysis of the sample, besides, was done using EDS. The secondary and backscattered electrons were used in image forming while X-rays were used to identify and quantify the chemicals present at the selected test surface. From Figure 6, it can be seen that the graphene–chitin sample was made up of mostly carbon (43.44%) and oxygen (20.44%). The Minor element of Na was also detected, which was due to contamination during the handling of the film prior to the characterization, and this can be caused simply by skin contact of the handler with the film.

## 3. Experimental Setup

The experimental setup of the Erbium-doped fiber laser (EDFL) is shown in Figure 7. A 1.5 m long Erbium-doped fiber (EDF) was used as the gain medium. The setup also consisted of a 980/1550 nm wavelength division multiplexer (WDM), an isolator, the newly fabricated graphene as SA, and an 95/5 output coupler, arranged in a ring configuration. The core and cladding diameter of the EDF are 8 µm and 125 µm, respectively. The numerical aperture of the EDF is 0.16 and has Erbium ion absorptions of 45 dB/m at 1480 nm and 80 dB/m at 1530 nm. The EDF was pumped by a 980 nm laser diode (LD) via the WDM. The use of an isolator ensured unidirectional propagation of the oscillating laser. The output of the laser was tapped from the cavity through a 95/5 coupler while keeping 95% of the light to oscillate in the ring cavity. The spectrum of the EDFL was inspected by using the optical spectrum analyzer (OSA) (Yokogawa AG6370B, Tokyo, Japan) with a spectral resolution of 0.05 nm, whereas the mixed domain oscilloscope (OSC) (Tektronix MDO3024, Beaverton, OR, USA) was used to observe the output pulse train and the signal-to-noise ratio (SNR) via a 460 kHz bandwidth photo-detector (PD) (Thorlab DET01CFC, Newtown, NJ, USA). The fabricated graphene–chitin film was placed in between two fiber ferrules with the aid of an index-matching gel before its integration in the fiber laser cavity.

## 4. Results and Discussions

A wavelength shift of around 7.05 nm and a widening of the spectral band of approximately 1.03 nm was observed upon the insertion of the graphene-chitin SA, as shown in Figure 8, which can be attributed to the insertion loss. The change in the refractive index of the graphene–chitin as it transitions from a low to a high intensity state can also cause the wavelength shift. At higher input pump power, the SA saturates, resulting in the decrease of its refractive index, causing a shorter resonance wavelength, as observed in Figure 8. Additionally, the broader spectrum might result from the nonlinearity of the SA itself, causing a degree of loss to the cavity upon its insertion [34,35].

The input pump power was then increased steadily, recording a Q-switched operation as the pump power reached 80.63 mW, and it remained steady up until 163.16 mW, with no damage inflicted on the film, indicating that the film was able to withstand high input power and can self-start a Q-switched lasing at low power.

Typical Q-switched laser output train and its single envelope under the maximum input pump power of 163.16 mW is shown in Figure 9a,b. The repetition rate of the Q-switched pulse train was 88.97 kHz, with corresponding shortest pulse duration of 1.54 μs and pulse-to-pulse separation of 11 μs. The variations of the Q-switching pulses with the increasing pump power were observed and recorded in Figure 10a,b. Increasing the input pump power from 80.63 mW to 163.16 mW caused the repetition rate to increase from 14.49 kHz to 88.97 kHz while the pulse width was reduced from 12.08 μs to 1.54 μs, which resulted from the increase of the pump rate of the upper laser level.

The pulse width obtained was lower than many of the previously reported single-regime Q-switched operation using graphene-PVA based Sas, and it was comparable to the one employing a graphene-silica hybrid waveguide and optically deposited graphene oxide [32,33,34]. It was also lower than those reported using MXenes and MAX phase [35,36,37,38,39]. The calculated instantaneous peak power and pulse energy was observed to follow the upward trend in which the peak power and the pulse energy maxed out at 53.3 mW and 82.08 nJ, respectively. The recorded pulse energy is higher than the one reported using SWCNT SA via evanescent field interaction as well as MXenes and MAX phases [35,36,37,38,39]. A high SNR of 57 dB was obtained at the maximum repetition rate of 88.97 kHz, as shown in Figure 11, wherein no changes or drifts were observed in the spectral widths and frequency stability, indicating high laser operation stability. This is supported further by a long-term stability test of over 6 h, from which the laser spectra and the pulse train were monitored every 1 h. With steady output intensity and no changes to the central wavelength of the output spectrum, a highly stable laser operation was achieved. This can be seen in the 6 h long observation of the optical spectrum and pulse train in Figure 12a,b, respectively in which the evolution at each hour represented by the different colors showed no obvious modulations, from hour one (purple) to hour 6 (orange).

As the pump power increased further than 163.16 mW, the pulse train observed in the oscilloscope began to become unstable, and by the point that the pump power reached 167.75 mW, a mode-locking operation can be seen to start taking over, though with a very unstable operation. At a high pumping level, the thermal accumulation and supersaturation of the SA produced a sharp decrease in the transmission of the laser pulse passing through the SA, leading to the generation of short pulses [9,40]. Eventually, at 172.33 mW, a self-started stable mode-locked operation was obtained due to energy quantization, with no need for polarization state adjustment. Graphene-based SA can be used for both Q-switching and mode-locking operations. During Q-switching, the graphene SA is operated in a nonlinear absorption regime, while during mode-locking, it is operated In a linear absorption regime. Graphene’s ability to generate electron-hole pairs and shift the Fermi level relative to the Dirac point is responsible for both its nonlinear and its linear absorption behavior. Graphene’s fast response time and strong nonlinear absorption make it a good switch, while its linear absorption response makes it a good modulator for inducing mode locking. The increase in pump power and matching round-trip time of the laser pulse and the compensated fiber dispersion can drive the laser cavity into a regime of high energy and short pulses, leading to fast and phase-locked emission. Polarization controllers, although helpful in controlling the polarization of the laser output, and, thus, helping to obtain the mode-locked operation in multiple works, does add to the complexity of the fiber laser system and increase the system’s insertion loss. The limited range of adjustment of the polarization controller also limits its usefulness in certain applications [41,42,43]. The optical spectrum of the mode-locked pulse is depicted in Figure 13, with the resolution of the OSA set at 0.02 nm. The central wavelength of the spectrum was at 1562.23 nm, with a wide 3 dB bandwidth measured at approximately 10.26 nm.

The central wavelength slightly shifted from the continuous wave (CW) of the laser cavity which was initially at 1564.48 nm, mainly due to the loss introduced by the insertion of the SA. Several pairs of Kelly sidebands can be observed from the spectrum, indicating a clear characteristic of a traditional negative dispersion soliton wherein the group velocity of the signal in the fiber is higher at a shorter wavelength. This resulted in a stable pulse that maintained its shape and its amplitude as it propagated through the fiber due to the balance between non-linear effects and dispersion. With high peak power as one of its characteristics, this explains why the peak power of several magnitudes is higher than that of Q-switching operation [44,45,46], and this is a necessity in optical communication where efficient data transmission requires high power.

The mode-locked pulse was observed to be stable from the threshold pump power of 172.33 mW to the maximum pump power of 181.50 mW. The usual limitation of film-based SA in withstanding high input power [47,48] was not observed in this study due to the high thermal damage of graphene and chitin, which have higher thermal stability when compared to conventional polymers such as PEO and PVA [49,50,51]. Additionally, exposed to higher temperature, the harmful degradation products of synthetic polymers, such as toxic gases, can be avoided as chitin when degraded and will break down into chitosan, which has been reported to happen at a temperature between 200 °C and 300 °C [52,53]. The pulse train is presented in Figure 14a with a uniform pulse-to-pulse interval of approximately 38 ns, which was consistent with the cavity round-trip time. The fundamental frequency of 26.11 MHz was found to be stable at the mode-locking state while concurring to the total cavity length of around 8 m. Figure 14b shows the autocorrelation trace at the pump power of 181.50 mW. By applying sech^2^ fitting, the pulse duration at its FWHM was estimated to be about 0.70 ps, resulting in a pulse duration of 1.08 ps (τ_FWHM_/1.54). The autocorrelation trace revealed that the result of the experiment follows the sech^2^ fitting closely. The calculated output power and pulse energy of the mode-locked EDFL was 39.75 W and 0.043 nJ, respectively.

The RF spectrum of the generated mode-locked pulse was recorded with a span of 500 MHz and 50 MHz, respectively, as shown in Figure 15a,b. A sharp signal at 26.11 MHz, corresponding to the fundamental frequency of the laser cavity, was observed. The background noise was suppressed by 40 dB from the peak signal, highlighting low-amplitude noise fluctuation and good mode-locking stability. The wideband RF spectrum was up to 500 MHz, where an evenly spaced frequency interval was observed, and which was free from spectral modulation. Figure 16a,b shows the 6-h stability test of the optical spectrum analyser (OSA) and oscilloscope trace for each hour represented by the different colors for each hour, indicating stable mode-locking operation with consistent −3 dB linewidths and no timing jitter. A consistent 3 dB bandwidth as demonstrated here is useful for applications requiring stable laser performance such as optical communications and sensing system which needs efficient transmission of signal over long distances without distortion or loss of information. A comparison of the obtained mode-locked operation to previously reported ones using graphene-based SAs are summarized in Table 1. This work has the advantage of dual-regime operation in a single laser cavity, and, also, mode-locking without an additional component, such as a polarization controller or a single-mode fiber (SMF).

## 5. Conclusions

Graphene–chitin film-based SA was used to generate simultaneous Q-switching and mode-locking operations in the telecommunication region. The lasing produced a Q-switched 1.54 μs pulse at a high input pump power of 163.16 mW. This operation was able to produce a high peak power and a pulse energy of 53.3 mW and 82.08 nJ, respectively. Beyond the input pump power of 163.16 mW, the pulse started to become unstable, and upon reaching 172.33 mW, the mode-locked pulse can be observed. The 26.11 MHz pulse correlated to a 1.08 ps pulse width with a peak power of 39.75 W. Although progressive works are still needed to improve the performance of the Q-switched and the mode-locked EDFL––such as by optimizing the cavity configuration through output coupling efficiency, by gaining medium doping concentration, and by improving the SA quality––this finding will be of interest due to their application in material processing, optical spectroscopy, and telecommunication.

## Figures and Tables

**Figure 1 micromachines-14-01048-f001:**
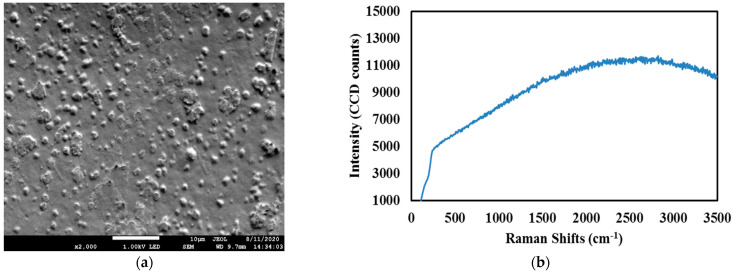
FESEM images and Raman spectrum of (**a**,**b**) pure chitin film, (**c**,**d**) graphene filament slurry, and (**e**,**f**) graphene-chitin film. Adapted with permission from Elsevier [23].

**Figure 2 micromachines-14-01048-f002:**
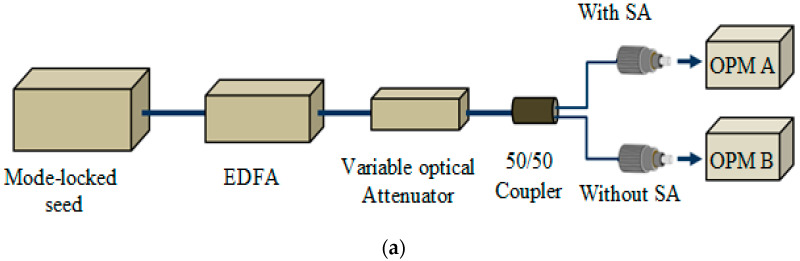
Nonlinear measurement of graphene–chitin film: (**a**) twin balanced detector setup, (**b**) modulation depth measurement.

**Figure 3 micromachines-14-01048-f003:**
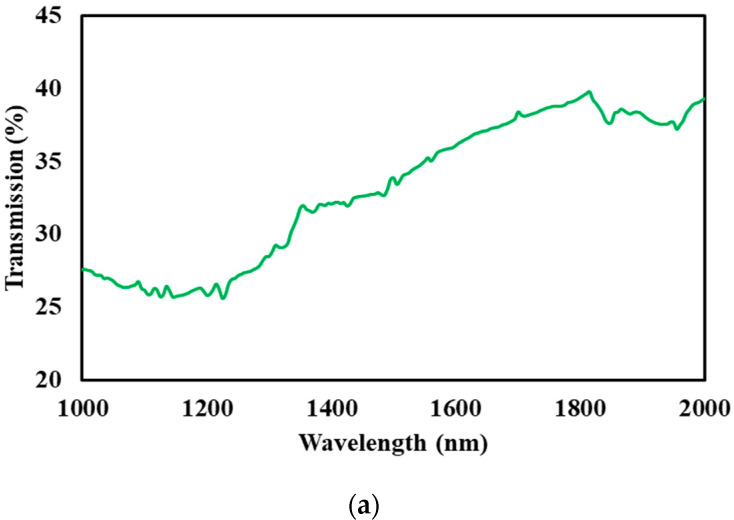
UV-Vis-NIR (**a**) transmission and (**b**) absorption spectrum of graphene-chitin.

**Figure 4 micromachines-14-01048-f004:**
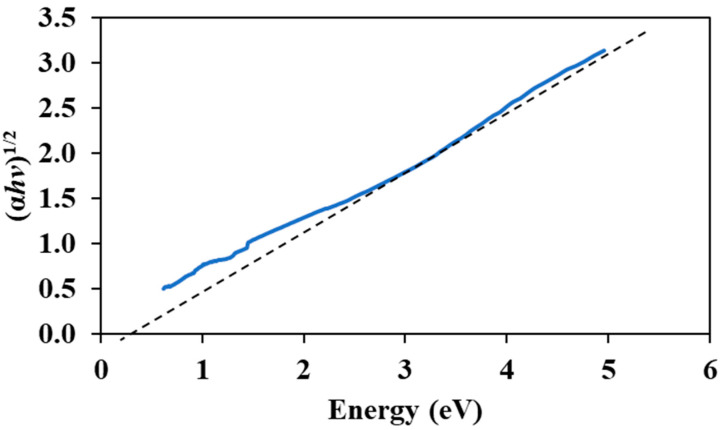
Tauc plot of graphene-chitin film.

**Figure 5 micromachines-14-01048-f005:**
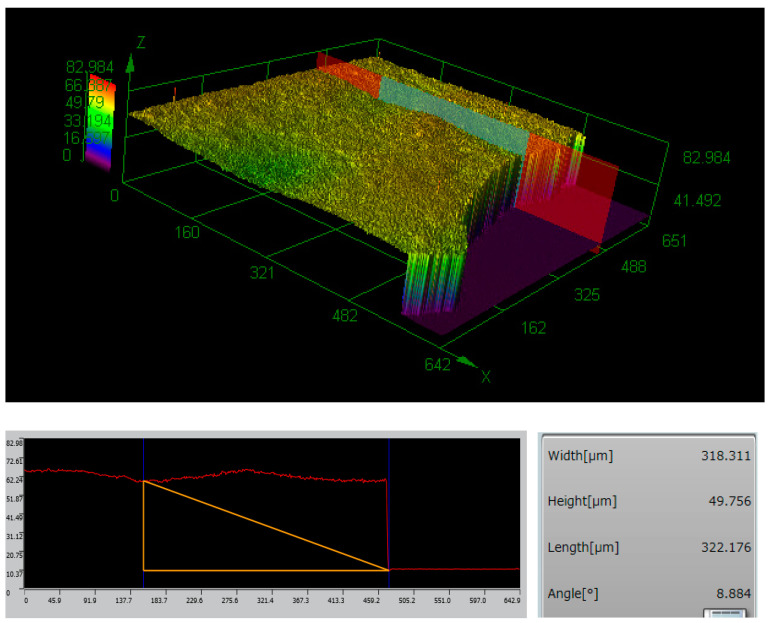
Thickness measurement of the graphene-chitin film.

**Figure 6 micromachines-14-01048-f006:**
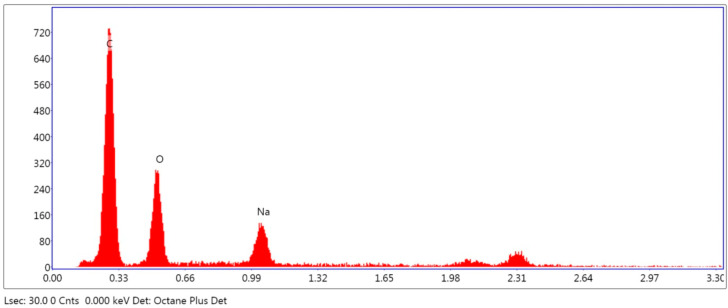
Chemical composition of graphene-chitin film.

**Figure 7 micromachines-14-01048-f007:**
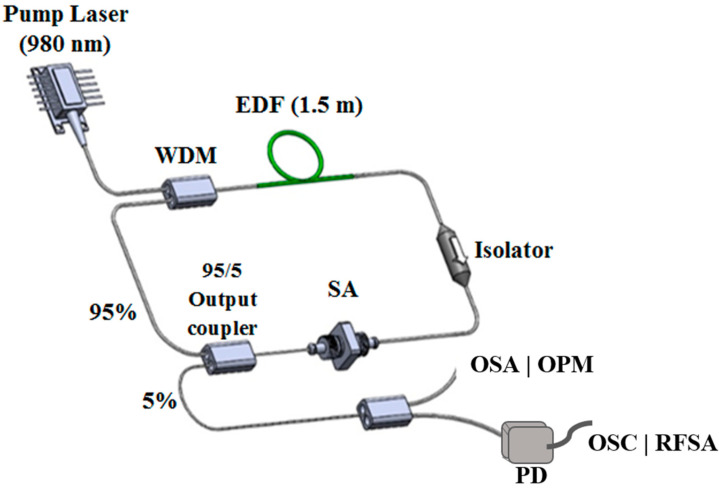
Experimental setup of EDFL incorporating graphene-chitin film-based SA.

**Figure 8 micromachines-14-01048-f008:**
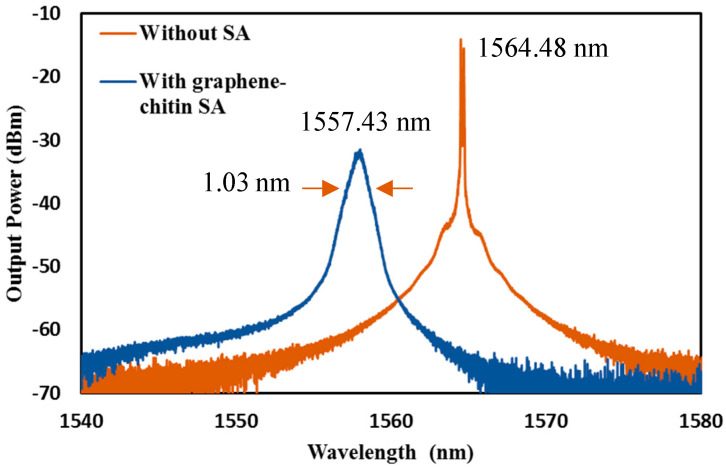
Optical spectrum of the EDFL before and after graphene-chitin SA integration.

**Figure 9 micromachines-14-01048-f009:**
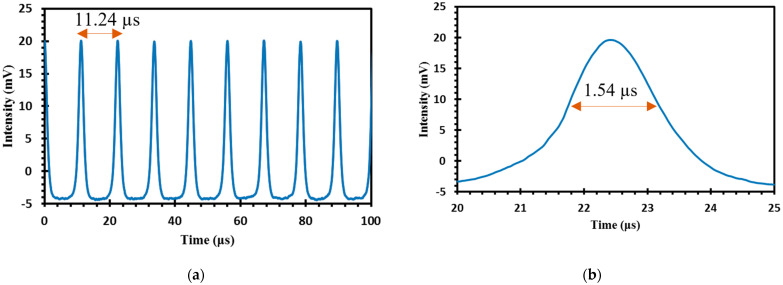
Q-switched pulsed train of the graphene-chitin based EDFL: (**a**) OSC trace with pulse separation of 11.24 µs; (**b**) Single pulse envelope with a pulse width of 1.54 µs.

**Figure 10 micromachines-14-01048-f010:**
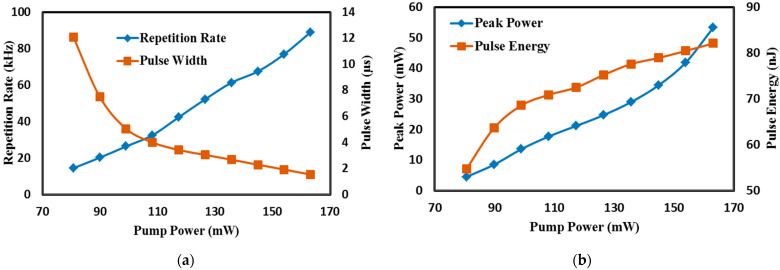
Q-switched pulsed train trend of the graphene-chitin based EDFL: (**a**) Repetition rate and pulse width vs the input pump power; (**b**) Peak power and pulse energy vs the input pump power.

**Figure 11 micromachines-14-01048-f011:**
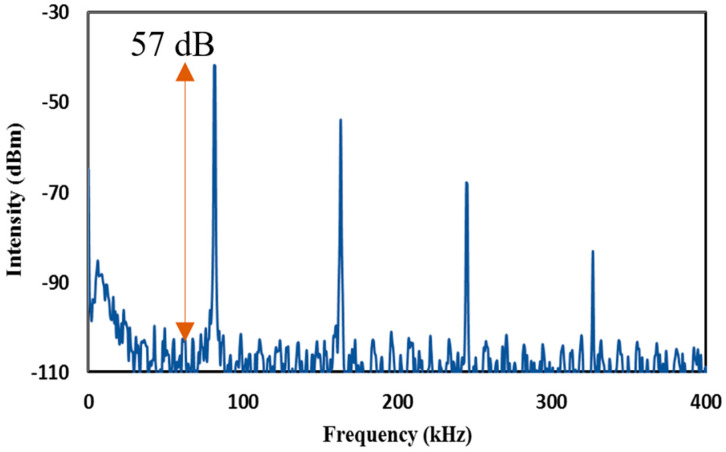
RFSA measurement of the graphene-chitin film based EDFL at SNR of 57 dB.

**Figure 12 micromachines-14-01048-f012:**
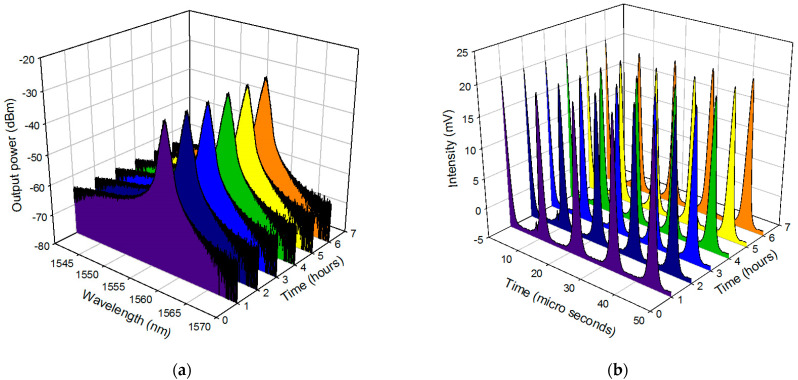
The stability of the output spectrum and pulse train for graphene-chitin based SA as passive Q-switcher over 6 h at a 1-h interval.

**Figure 13 micromachines-14-01048-f013:**
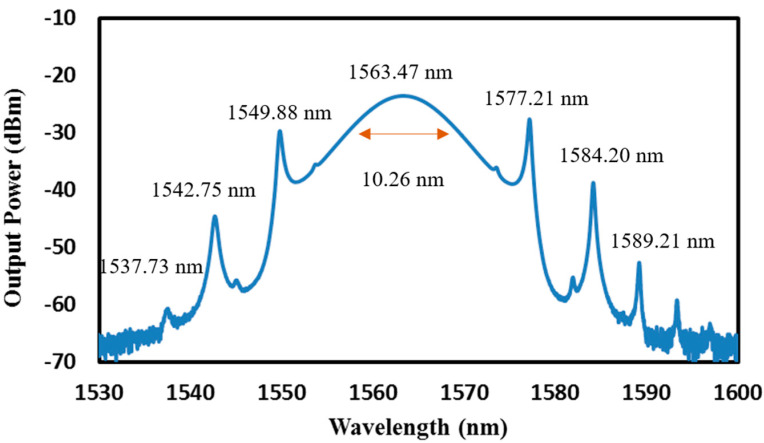
Optical spectrum of the soliton mode-locked EDFL based on graphene–chitin film.

**Figure 14 micromachines-14-01048-f014:**
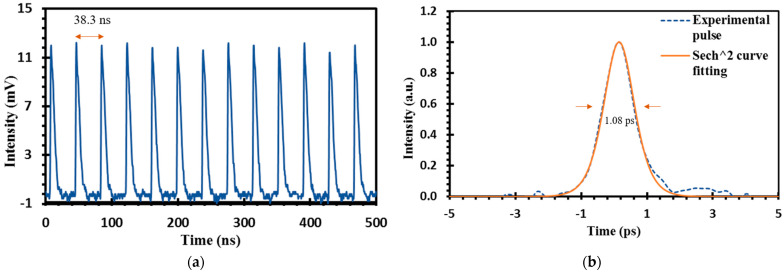
(**a**) Mode-locked pulse trace with pulse separation of 38.3 ns; (**b**) Autocorrelation trace with pulse width of 1.08 ps.

**Figure 15 micromachines-14-01048-f015:**
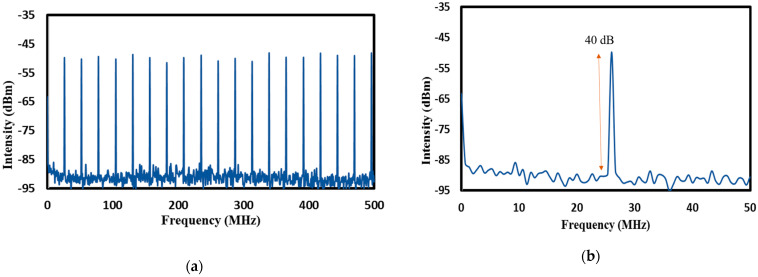
RFSA measurement of the mode-locked EDFL based on graphene-chitin: (**a**) RF spectrum at 500 MHz span; (**b**) RF spectrum at 50 MHz span showing an SNR of 40 dB.

**Figure 16 micromachines-14-01048-f016:**
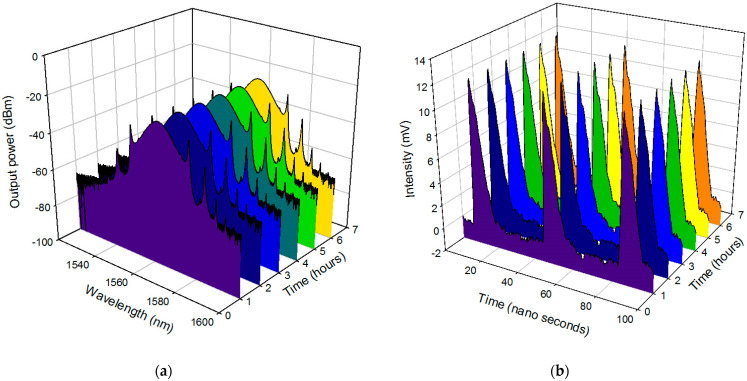
The stability of the output spectrum for graphene-chitin based SA as passive mode locker over 6 h at a 1-h interval.

**Table 1 micromachines-14-01048-t001:** Mode-locked EDFL using graphene based SA in the 1.5 µm region.

Material	Center WaveLength (nm)	Threshold Pump Power (mW)	Pulse Width	Repetition Rate (MHz)	SNR (dB)	Modulation Depth (%)	Average Output Power (mW)	Ref
Graphene Nanoplatelets	1559.26	86.3	0.59 ns	21.36	-	27.7	-	[54]
Graphene–DNA	1563	280	0.82 ps	14.11	50	-	7.5	[55]
Graphene Nanoplatelets	1558.35	22.6	0.69 ps	13.11	58.2	-	6.7	[56]
GO-PEO	1558.6	70	1.25 ps	21.8	-	-	0.363	[57]
rGO	1567.29	273	1.38 ns	12.66	50	5.5	-	[58]
CVD graphene–PMMA	1569.5	93	24 ns	5.78	65	-	12.1	[59]
CVD Graphene–PMMA	1555	100	0.252 ps	56.37	-	-	15.66	[60]
Graphene–PMMA	1562.7	45	0.967 ps	14.3	61.3	1.52	-	[61]
Graphene–chitin	1563.47	172.33	1.08 ps	26.11	40	15.08	53.3	This work

## Data Availability

Not applicable.

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
