# Peer review of "Dual Regime Mode-Locked and Q-Switched Erbium-Doped Fiber Laser by Employing Graphene Filament–Chitin Film-Based Passive Saturable Absorber"

_micromachines, 2023, doi:10.3390/mi14051048_

Round 1

Reviewer 1 Report

In the manuscript, authors investigate the dynamics of high energy dual regime unidirectional Erbium-doped fiber laser in ring cavity which is passively Q-switched and mode-locked through the use of an environmentally friendly graphene filament-chitin film based saturable absorber. My opinion is that the paper has some interest and deserve the publication in Micromachines after the following concerns are appropriately responded.

1.       In the article, the preparation process and analysis of characterization of graphene filament-chitin film is too brief, only FESEM and Raman spectrum are not enough for characterizing this SA, I suggest the authors provide thorough analysis of this as-prepared sample to illustrate its specific advantage as SA.

2.       In the article, the explanation of the wavelength shift between CW and PQS is not convincible, author should give more detailed explanation, and authors should provide evidence about how does the nonlinearity of the SA contributes to the change in the wavelength during the PQS mode operation.

3.       In figure 8, how to demonstrate high laser operation stability?

The quality of English language need to be further improved.

Reviewer 2 Report

The authors demonstrated Q-switched and mode-locked pulses operation from fiber laser based on the graphene filament-chitin film SA. I have some concerns over the manuscript:

1. Could the author explain why the Q-switched and mode-locked pulses can be acquired in the same cavity? What are the states of the SA in these two lasing states? What is the motivation to try this material as SA?

2. How about the long-term stability of the SA in the Q-switched and mode-locked states? Please specify in the manuscript with experimental data.

3. What about the damage threshold of the SA and what about the performance uniformity of the SA?

4. From Fig.2b, the absorption(loss) of the SA is very high (no less than 85%), but why the slope efficiency in Fig.7b is such high(more than 50%)? Is there any contradiction?

The quality of the English language is fine and acceptable.

Reviewer 3 Report

I think that this paper is suitable for publication in this Journal. But I suggest some minor issues. Please check the attached file

Round 2

Reviewer 1 Report

No further comments

Reviewer 2 Report

I am satisfied with the revised manuscript and now I think it can be accepted for publication.